# Correlation Analysis of Microbial Contamination and Alkaline Phosphatase Activity in Raw Milk and Dairy Products

**DOI:** 10.3390/ijerph20031825

**Published:** 2023-01-19

**Authors:** Zixin Peng, Ying Li, Lin Yan, Shuran Yang, Dajin Yang

**Affiliations:** Research Unit of Food Safety, Chinese Academy of Medical Science (2019RU014), NHC Key Laborarory of Food Safety Risk Assessment, China National Center for Food Safety Risk Assessment, Beijing 100022, China

**Keywords:** microbial contamination, alkaline phosphatase activity, raw milk, pasteurized milk, sterilized milk, aerobic *Bacillus*, thermophilic aerobic *Bacillus*

## Abstract

Microbial contamination in raw milk and dairy products can detrimentally affect product quality and human health. In this study, the aerobic plate count, aerobic *Bacillus* abundance, thermophilic aerobic *Bacillus* abundance, and alkaline phosphatase activity were determined in 435 raw milk, 451 pasteurized milk, and 617 sterilized milk samples collected from 13 Chinese provinces (or municipalities). Approximately 9.89% and 2.22% of raw milk and pasteurized milk samples exceeded the threshold values for the aerobic plate count, respectively. The proportions of aerobic *Bacillus* in raw milk, pasteurized milk, and sterilized milk were 54.02%, 14.41%, and 1.30%, respectively. The proportions of thermophilic aerobic *Bacillus* species were 7.36% in raw milk and 4.88% in pasteurized milk samples, and no bacteria were counted in sterilized milk. Approximately 36.18% of raw milk samples contained >500,000 mU/L of alkaline phosphatase activity, while 9.71% of pasteurized milk samples contained >350 mU/L. For raw milk, there was a positive correlation between the aerobic plate count, the aerobic *Bacillus* abundance, and the alkaline phosphatase activity, and there was a positive correlation between the aerobic *Bacillus* abundance, the thermophilic aerobic *Bacillus* count, and the alkaline phosphatase activity. For pasteurized milk, there was a positive correlation between the aerobic plate count, the aerobic *Bacillus* abundance, and the thermophilic aerobic *Bacillus* count; however, the alkaline phosphatase activity had a negative correlation with the aerobic plate count, the aerobic *Bacillus* abundance, and the thermophilic aerobic *Bacillus* abundance. These results facilitate the awareness of public health safety issues and the involvement of dairy product regulatory agencies in China.

## 1. Introduction

Microbial contamination of raw milk and dairy products is an important source of foodborne pathogens that can adversely affect human health [1,2]. The microbial contamination of raw milk can affect the safety and the quality of dairy products from the source [3]. During the processing of dairy products, microorganisms from several sources (e.g., personnel, water, equipment, additives, and packaging materials) can cause contamination [4]. Milk microbial contamination is also responsible for significant economic losses at various points throughout the milk production chain [5]. Compared with other commodities, dairy products are easily contaminated and subject to rapid deterioration [4].

The most common bacterial spores in dairy products belong to the genus *Bacillus*. Mesophilic and thermophilic aerobic *Bacillus* species are of particular concern because of their high heat resistance and the high thermal stability of their degradation enzymes [6]. Some species of the genus *Bacillus* have been implicated in food-borne diseases. For example, *B. cereus* was reported as the causative agent in a large food poisoning outbreak attributed to pasteurized milk [7]. At mesophilic temperatures, some facultative thermophiles, such as *B. subtilis*, *B. licheniformis*, and *B. pumilus*, can also produce toxins [8]. Spoilage caused by *Bacillus* species has been reported, even in commercially sterilized milk [7].

Alkaline phosphatase (ALP; EC 3.1.3.1), an enzyme that is naturally found in raw milk, can be denatured by pasteurization temperatures. Alkaline phosphatase activity has been used to confirm the efficacy of pasteurization in dairy products. Thus, a dairy product that contains an insignificant amount of active enzyme or no enzyme at all is considered properly pasteurized [9].

The aims of this study were: (i) to investigate microbial contamination and alkaline phosphatase activity in raw milk, pasteurized milk, and sterilized milk collected from 13 Chinese provinces, and (ii) to clarify the correlation between the aerobic plate count, aerobic *Bacillus* abundance, thermophilic aerobic *Bacillus* abundance, and alkaline phosphatase activity. The findings of this study provide theoretical and practical support for elucidating the microbiological quality of raw, pasteurized, and sterilized milk in China.

## 2. Materials and Methods

### 2.1. Sampling 

A total of 435 raw milk, 451 pasteurized milk, and 617 sterilized milk samples were collected randomly from 13 Chinese provinces (or municipalities). One province collected 105–189 samples (Appendix A), consisting of raw milk, pasteurized milk, and sterilized milk samples. The volume of each sample was 500 mL. The sampling sites included dairy farms, dairy factories, supermarkets, retail shops, online stores, farmer’s markets, and restaurants. The sample collection and the investigation were conducted from April to September 2021.

### 2.2. Microbiological Analyses and Enzymatic Activity Assays

The determination of the aerobic plate count was conducted following the Chinese national food safety standard [GB 4789.2-2016] [10]. Briefly, a 25 g raw milk or dairy product sample was added into 225 mL of 0.85% saline solution and homogenized for 2 min to prepare a 10^−1^ dilution. Serial dilutions were prepared with 0.85% saline solution. Three suitable serial dilutions were selected according to the contamination status of the samples. Next, 1 mL was aliquoted from each diluent, transferred into agar medium, and incubated for 48 h at 36 °C. All the colonies appearing on the plates were enumerated. The data were reported as log CFU/g of raw milk or dairy product. The limit of detection (LOD) of this method was 1 log_10_ CFU/mL.

The plate counts of aerobic *Bacillus* and thermophilic aerobic *Bacillus* were determined according to the methods of NEN 6813:2014 [11] and NEN 6809:2014 [12].

For aerobic *Bacillus* detection, a 25 g raw milk or dairy product sample was added into 225 mL of phosphate buffer and homogenized for 2 min to prepare a 10^−1^ dilution. A 10 mL diluted sample was transferred into a sterile tube, which was incubated in an 80 °C water bath for 10 min and then cooled in a 20 °C freezer. The cooled sample was diluted serially with phosphate buffer. Three suitable serial dilutions were selected according to the contamination status of the samples. Next, 1 mL was aliquoted from each diluent, transferred into milk plate count (MPC) agar medium, and incubated for 48 h at 36 °C. All the colonies appearing on the plates were enumerated. The data were reported as log CFU/g of raw milk or dairy product. The LOD of this method was 1 log_10_ CFU/mL.

For thermophilic aerobic *Bacillus* detection, a 25 g raw milk or dairy product sample was added into 225 mL of phosphate buffer and homogenized for 2 min to prepare a 10^−1^ dilution. A 10 mL diluted sample was transferred into a sterile tube, which was incubated in a 100 °C water bath for 30 min and then cooled in a 20 °C freezer. The cooled sample was diluted serially with phosphate buffer. Three suitable serial dilutions were selected according to the contamination status of the samples. Next, 1 mL was aliquoted from each diluent, transferred into dextrose tryptone agar (DTA) medium, and incubated for 48 h at 36 °C. All the colonies appearing on the plates were enumerated. The data were reported as log CFU/g of raw milk or dairy product. The LOD of this method was 1 log_10_ CFU/mL.

The alkaline phosphatase activity in raw milk and pasteurized milk was measured according to a modified chemiluminescent method [13]. Briefly, a 100 μL milk sample was added into a vial containing 0.5 mL of predispensed chemiluminescent substrate (Beijing Biotai, China) in buffer. The contents in the vial were mixed for 5 s, and the vial was attached to a NovaLUM adapter (Charm Sciences, Lawrence, MA, USA) and inserted upright into a NovaLUM analyzer. The fast alkaline phosphatase (F-AP) channel specific to the matrix in the NovaLUM analyzer was activated. The F-AP channel was equipped with a built-in timer and temperature monitor to complete the analysis in 45 s for raw milk or dairy product samples. The LOD of this method was 1.30 log_10_ mU/L (20 mU/L), and the LOQ was 1.78 log_10_ mU/L (60 mU/L).

### 2.3. Statistical Analysis

The mean, standard deviation, median, minimum and maximum values, and 25th and 75th percentiles were calculated for each microbiological parameter using SPSS 16.0 software (IBM, Armonk, NY, USA) [14]. Spearman’s correlation analysis was conducted using R, and a *p* value < 0.05 was indicative of a significant correlation. The values of the aerobic plate count, aerobic *Bacillus* count, and thermophilic aerobic *Bacillus* count <LOD were considered as 0, while the values of alkaline phosphatase activity <LOQ were also considered as 0. The correlation graph was produced using an online tool (http://www.bioinformatics.com.cn/plot_basic_corrplot_corrlation_plot_082, (accessed on 7 January 2023)) [15]. 

## 3. Results

### 3.1. Microbiological and Enzymatic Activity Analyses

The results of the microbiological and enzymatic activity analyses of raw and pasteurized milk are outlined in Table 1, and the results of sterilized milk are outlined in Table 2.

The contamination levels of the raw milk samples varied widely. The mean and median values of the aerobic plate count were far below the threshold set by the Chinese national food safety standard [GB19301-2010] (6.30 log_10_ CFU/mL). Approximately 9.89% (43/435) of samples had a contamination level higher than the threshold value. The proportions of aerobic *Bacillus* and thermophilic aerobic *Bacillus* were 54.02% (235/435) and 7.36% (32/435) in raw milk samples, respectively. Approximately 36.18% (157/434) of raw milk samples contained >500,000 mU/L (5.70 log_10_ mU/L) alkaline phosphatase activity.

For the pasteurized milk samples, the mean and median values of the aerobic plate count were far below the threshold (5.00 log_10_ CFU/mL). However, 2.22% (10/451) of samples showed a contamination level that was higher than the threshold value, indicating the low hygienic status of milk. The proportions of aerobic *Bacillus* and thermophilic aerobic *Bacillus* were 14.41% (65/451) and 4.88% (22/451) in pasteurized milk samples, respectively. The maximum values were 5.30 log_10_ CFU/mL and 3.81 log_10_ CFU/mL for the counts of aerobic *Bacillus* and thermophilic aerobic *Bacillus*, respectively. Approximately 9.71% (43/443) of pasteurized milk samples contained >350 mU/L (2.54 log_10_ CFU/mL) of alkaline phosphatase activity. Of the 43 samples, only 2 (4.65%) samples showed aerobic *Bacillus* count >1 log_10_ CFU/mL, and 1 (2.33%) sample showed thermophilic aerobic *Bacillus* above the LOD. 

The contamination levels of the sterilized milk samples were very low. The proportion of aerobic *Bacillus* was 4.05% (25/617). The contamination of aerobic *Bacillus* in sterilized milk was 1.30% (8/617). The maximum values were 2.32 log_10_ CFU/mL and 1.34 log_10_ CFU/mL for the aerobic plate count and the aerobic *Bacillus* count, respectively. No thermophilic aerobic *Bacillus* was counted in the samples.

### 3.2. Correlation Analysis of Microbial Contamination and Alkaline Phosphatase Activity

The correlogram based on the Spearman correlation analysis revealed the influence of the aerobic plate count, counts of aerobic *Bacillus* and thermophilic aerobic *Bacillus*, alkaline phosphatase activity after comparing one to another for raw milk (Figure 1) and pasteurized milk (Figure 2), respectively.

For raw milk, there were positive correlations between the aerobic plate count, the aerobic *Bacillus* abundance, and the alkaline phosphatase activity (all *p* < 0.05), and the aerobic *Bacillus* abundance had positive correlations with the thermophilic aerobic *Bacillus* count and the alkaline phosphatase activity (both *p* < 0.05). Thermophilic aerobic *Bacillus* count was irrelevant in raw milk in the determination of correlations with the aerobic plate count and the alkaline phosphatase activity, respectively (both *p* > 0.05). The largest *p* value (0.29) was observed for the positive correlation between the aerobic plate count and the aerobic *Bacillus* abundance.

For pasteurized milk, there were positive correlations between the aerobic plate count, the aerobic *Bacillus* abundance, and the thermophilic aerobic *Bacillus* count (all *p* < 0.05); however, the alkaline phosphatase activity had negative correlations with the aerobic plate count, the aerobic *Bacillus* abundance, and the thermophilic aerobic *Bacillus* abundance (all *p* > 0.05). Similar to raw milk, the largest *p* value (0.36) was observed for the positive correlation between the aerobic plate count and the aerobic *Bacillus* abundance.

## 4. Discussion

Raw milk safety is crucial for both farmers’ income and human health, as well as for consumers paying more for safer dairy products [18]. Microbial contamination of raw milk originating from farms can increase spoilage and wastage and adversely affect producers, traders, and consumers [19]. The aerobic plate count is an important criterion for evaluating the microbial quality of raw milk as well as the degree of food freshness [20]. This criterion reflects the standards of primary operation procedures that include collection, transportation, and storage [18]. In our survey, nearly 10% of raw milk samples exceeded the Chinese legal limit (≤2 × 10^6^ CFU/ mL). Although the legal limit is far below those of the EU and USA (<1 × 10^5^ CFU/mL) [20], these results indicate that more attention should be given at the farm level, as most microorganisms are introduced into final dairy products at this stage. 

Pasteurization does not have a significant impact on the nutritional value of raw milk [21]. This process is essential for ensuring the safety of milk and increasing its shelf life, because it reduces most heat-resistant and all other non-spore-forming microbes to safer levels, thereby increasing the safety and shelf-life. Factors affecting the shelf-life of pasteurized milk include storage temperature, post-pasteurization contamination, growth behaviors of contaminating bacteria, and incidence of *Bacillus* occurrence [22]. In most countries, the legal limits for aerobic plate count in pasteurized milk range from 5 × 10^3^ to 5 × 10^5^ CFU/mL [23]. According to the legal limit of China (<1 × 10^5^ CFU/mL), approximately 2.22% of samples exceeded this threshold value, indicating the low hygienic status of milk. The acceptance limit for the aerobic plate count in pasteurized milk is ≤2 × 10^4^ CFU/mL [23]. Although milk pasteurization is regarded as an effective method to eliminate foodborne pathogens, some reports on pathogen contamination in pasteurized milk clearly indicate that pasteurization alone is not the ultimate solution for the control of milk-borne pathogens [22,24,25]. However, pasteurization is still an optimized method that minimizes bacterial contamination and maintains high nutritive value [17].

Sterilized milk is created by heating milk through an ultra-high temperature process. This process can destroy nearly all microbes in milk, thereby increasing the shelf-life [26]. To achieve the legal microbe limit for sterilized milk in China (GB25190-2010), milk must be sterilized [27]. In this survey, only 4% of sterilized milk samples showed an aerobic plate count ≥1 log_10_ CFU/mL. A previous study demonstrated that most microorganisms present in sterilized milk were heat-treatment-resistant strains or those that originated from post-sterilization contamination [23]. 

The genus *Bacillus* is capable of overcoming the heat barrier during the sterilization of milk [28]. Some of these microorganisms can produce highly heat-resistant endospores, which may survive the ultra-high temperature process [6]. The aerobic *Bacillus* could not only affect the shelf-life of pasteurized and sterilized milk but is also associated with defects such as off flavors, sweet curdling, and bitter cream [6]. A previous study demonstrated that soiling on the udder and the teats is the major source of aerobic *Bacillus* contamination [8]. In this study, the proportions of aerobic *Bacillus* in raw milk, pasteurized milk, and sterilized milk were 54.02%, 14.41%, and 1.30%, respectively. Our results also showed that the proportions of thermophilic aerobic *Bacillus* were 7.36% and 4.88% in raw milk and pasteurized milk, respectively. A survey conducted in Tunisia found a high degree of diversity, both phenotypic and genotypic, among *Bacillus* isolates from milk samples. Seven *Bacillus* species, *Bacillus cereus* predominantly, were identified in pasteurized milk and sterilized milk and posed the risk of milk-borne illness to consumers [28]. Thus, it is important to minimize aerobic *Bacillus* contamination at the farm level. Moreover, our results showed that the aerobic plate count had a positive correlation with the aerobic *Bacillus* count in raw milk, and also had positive correlations with the aerobic *Bacillus* count and the thermophilic aerobic *Bacillus* in sterilized milk. Thus, good hygiene practices for the interior or the exterior of the udder and milking instruments during the milk production process must be implemented.

Alkaline phosphatase is a heat-sensitive enzyme found in raw milk that is used as a marker for the efficacy of thermal pasteurization. The absence of alkaline phosphatase activity has been used to confirm the benefits of pasteurization in dairy products for the past 80 years. A product that contains a small amount of active enzyme or no enzyme at all is considered properly pasteurized [9]. In our survey, a total of 36.18% of raw milk samples contained >500,000 mU/L alkaline phosphatase activity, which indicated widespread active enzyme. However, the value was decreased dramatically for pasteurized milk, with only 9.71% of samples containing >350 mU/L alkaline phosphatase activity based on EU guidelines (nos. 1664/2006 and 2074/2005), showing a good sterilizing effect. In the study of Ziobro and McElroy [9], nearly 14% (5/36) of pasteurized milk products showed >350 mU/L enzyme activity, which was higher than the value in this study. 

The thermal denaturation parameters for alkaline phosphatase activity in milk were similar to those of heat-resistant milk pathogens [9]. Our results showed that alkaline phosphatase activity showed weak positive correlations with the aerobic plate count and the aerobic *Bacillus* count in raw milk. By contrast, the alkaline phosphatase activity showed weak negative correlations with the aerobic plate count, the aerobic *Bacillus* count, and even the thermophilic aerobic *Bacillus* count in pasteurized milk. Alkaline phosphatase has greater heat resistance than most aerobic microorganisms. Therefore, most microorganisms will inactivate more rapidly than this enzyme during thermal degradation. Our results of alkaline phosphatase activity showed a weak negative correlation with the thermophilic aerobic *Bacillus* count, although these microorganisms have more thermal resistance. These results indicated that the alkaline phosphatase activity had a weak correlation with the microbial loads. Thus, enzyme activity is only an indicator of temperature and time of the heat treatment, and not an indicator of microorganisms present. A previous study indicated that the alkaline phosphatase activity assay was affected by many factors, including the composition of the product and the presence of microbial alkaline phosphatase [29]. However, our results showed that of the 43 (9.71%) pasteurized milk samples containing alkaline phosphatase activity >350 mU/L, only 2.22% samples showed aerobic plate count exceeded the legal limit (1 × 10^5^ CFU/mL) and 4.65% and 2.33% samples showed aerobic *Bacillus* count and thermophilic aerobic *Bacillus* > 1 log_10_ CFU/mL, respectively. These results illustrate that the alkaline phosphatase activity was a good indicator of the effectiveness of thermal pasteurization. Thus, detection and identification of the species of thermophilic aerobic *Bacillus* are necessary for the pasteurized milk samples with high alkaline phosphatase activity.

## 5. Conclusions

The output and consumption of dairy products in China have increased greatly in recent decades, and Chinese consumers have expressed increased demand for safe and healthy dairy products. Compared with other foods, dairy products are easily contaminated with microorganisms at farm, transportation, process, and storage stages and subject to rapid deterioration and foodborne diseases. Thus, further national-scale surveys of microbial contamination are needed for raw milk and dairy products. 

To our knowledge, this is the first study to evaluate the microbiological quality of raw milk and dairy products from 13 major dairy production and consumption provinces in China. Our results indicate that raw milk showed high microorganism contamination; however, the pasteurized milk and sterilized milk were sufficiently sterilized. In this study, we also improved our understanding about the correlation between microbial contamination and alkaline phosphatase activity. This study recommends that further studies be carried out to identify the contamination sources and the microorganism species contaminating pasteurized milk and sterilized milk.

## Figures and Tables

**Figure 1 ijerph-20-01825-f001:**
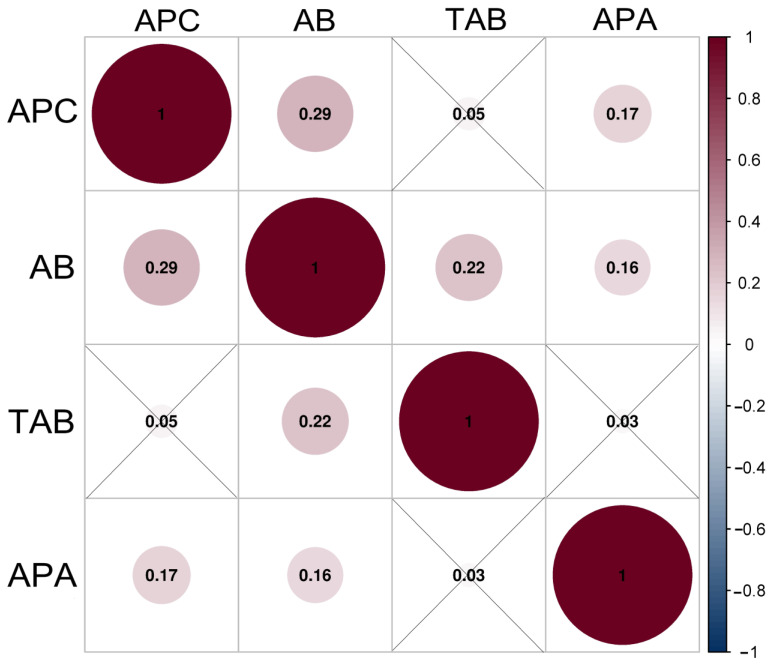
Correlation between the aerobic plate count, aerobic *Bacillus* count, thermophilic aerobic *Bacillus* count, and alkaline phosphatase activity of raw milk samples. The *p* values are shown in circles. The *p* value > 0.05 is shown in the crossover line, and the other *p* values are <0.05. APC: aerobic plate count; AB: aerobic *Bacillus* count; TAB: thermophilic aerobic *Bacillus* count; APA: alkaline phosphatase activity.

**Figure 2 ijerph-20-01825-f002:**
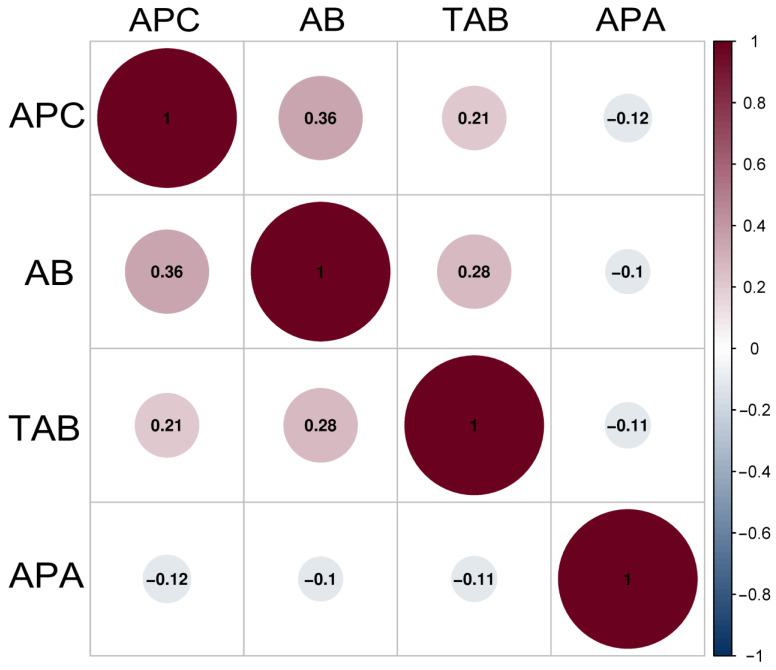
Correlation between the aerobic plate count, aerobic *Bacillus* count, thermophilic aerobic *Bacillus* count, and alkaline phosphatase activity of pasteurized milk samples. The *p* values are shown in circles. All the *p* values are <0.05. APC: aerobic plate count; AB: aerobic *Bacillus* count; TAB: thermophilic aerobic *Bacillus* count; APA: alkaline phosphatase activity.

**Table 1 ijerph-20-01825-t001:** Microbiological results and threshold values for raw and pasteurized milk.

Item *	Milk	Mean ± SD ^^^	Median	Min	Max	25th Perc.	75th Perc.	Threshold Values	Over-Limit Ratio
APC log_10_ CFU/mL	Raw	4.55 ± 1.43	4.37	<1	8.15	3.15	5.53	6.30 ^#^	9.89% (43/435)
	Pasteurized	2.83 ± 1.32	<1	<1	8.15	<1	2	5.00 ^&^	2.22% (10/451)
AB log_10_ CFU/mL	Raw	2.00 ± 0.82	<1	<1	5.04	<1	1.71	_	_
	Pasteurized	1.67 ± 0.76	<1	<1	5.30	<1	<1	_	_
TAB log_10_ CFU/mL	Raw	1.49 ± 0.44	<1	<1	2.48	<1	<1	_	_
	Pasteurized	1.61 ± 0.76	<1	<1	3.81	<1	<1	_	_
APA log_10_ mU/L	Raw	5.59 ± 0.40	5.65	1.96	6.68	5.51	5.82	_	_
	Pasteurized	2.22 ± 0.38	<1.78	<1.78	3.53	<1.78	2.08	2.54 ^@^	9.71% (43/443)

* APC: aerobic plate count; AB: aerobic *Bacillus* count; TAB: thermophilic aerobic *Bacillus* count; APA: alkaline phosphatase activity. ^^^ The mean ± SD only determined the value ≥LOD for APC, AB, and TAB or ≥LOQ for APA. ^#^ GB19301-2010 [16]. ^&^ GB19645-2010 [17]. ^@^ Commission Regulation (EC) No 1664/2006 of 6 November 2006 amending Regulation (EC) No 2074/2005 [13].

**Table 2 ijerph-20-01825-t002:** Microbiological results and threshold values for sterilized milk.

Item *	Mean ± SD ^^^	Median	Min	Max	25th Perc.	75th Perc.
APC log_10_ CFU/mL	1.45 ± 0.45	<1	<1	2.32	<1	<1
AB log_10_ CFU/mL	1.08 ± 0.15	<1	<1	1.34	<1	<1
TAB log_10_ CFU/mL	<1	<1	<1	<1	<1	<1

* APC: aerobic plate count; AB: aerobic *Bacillus* count; TAB: thermophilic aerobic *Bacillus* count. ^^^ The mean ± SD only determined the value ≥LOD for APC, AB, and TAB.

## Data Availability

Not applicable.

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
