# Peer review of "Correlation Analysis of Microbial Contamination and Alkaline Phosphatase Activity in Raw Milk and Dairy Products"

_ijerph, 2023, doi:10.3390/ijerph20031825_

Round 1

Reviewer 1 Report

Microbial contamination is a main potential risk in raw milk and dairy products, which could affect the quality and the safety of dairy products and even to be an important source of foodborne pathogens. This study determined the aerobic plate count, aerobic Bacillus abundance, thermophilic aerobic Bacillus abundance, and alkaline phosphatase activity in more than 1503 raw milk and dairy products collected from 13 main dairy production and consumption provinces in China. Through the surveillance, the authors indicated that raw milk showed high microorganism contamination but the pasteurized milk and sterilized milk were sufficiently sterilized. This study also showed a good positive correlation between the alkaline phosphatase activity and the thermophilic aerobic Bacillus count. This finding could be further used in developing a rapid technic to detect the thermophilic aerobic Bacillus contamination in pasteurized milk. The sample amount is large and has representative. This manuscript is well structured and easy to follow. From the longitudinal study, the microbial contamination in raw milk and dairy products in China can be compared and analysis. The results can give data support for the dairy production policies and regulations adjustment in China and even could give reference for worldwide, especially for developing countries. The language of this manuscript was carefully edited and free of grammatical error. The results could attract attention of the readers of IJERPH.

Author Response

Reviewer 1

Comments and Suggestions for Authors

Microbial contamination is a main potential risk in raw milk and dairy products, which could affect the quality and the safety of dairy products and even to be an important source of foodborne pathogens. This study determined the aerobic plate count, aerobic Bacillus abundance, thermophilic aerobic Bacillus abundance, and alkaline phosphatase activity in more than 1503 raw milk and dairy products collected from 13 main dairy production and consumption provinces in China. Through the surveillance, the authors indicated that raw milk showed high microorganism contamination but the pasteurized milk and sterilized milk were sufficiently sterilized. This study also showed a good positive correlation between the alkaline phosphatase activity and the thermophilic aerobic Bacillus count. This finding could be further used in developing a rapid technic to detect the thermophilic aerobic Bacillus contamination in pasteurized milk. The sample amount is large and has representative. This manuscript is well structured and easy to follow. From the longitudinal study, the microbial contamination in raw milk and dairy products in China can be compared and analysis. The results can give data support for the dairy production policies and regulations adjustment in China and even could give reference for worldwide, especially for developing countries. The language of this manuscript was carefully edited and free of grammatical error. The results could attract attention of the readers of IJERPH.

Dear Reviewer 1,

Thank you very much for your useful comments. Microbial contamination in raw milk and dairy products can detrimentally affect product quality and human health. During dairy processing, microorganisms from several sources, e.g., personnel, water, equipment, additives, and packaging materials, can contaminate the products. Compared with other foods, dairy products are easily contaminated with microorganisms and subject to rapid deterioration and foodborne diseases. Therefore, this work investigated microbial contamination and alkaline phosphatase activity longitudinally covering the raw milk and dairy products of 13 main milk production and consumption provinces. Moreover, this work analyzed the correlation among the aerobic plate count, aerobic Bacillus abundance, thermophilic aerobic Bacillus abundance, and alkaline phosphatase activity, in order to improve our understanding about the correlation between microbial contamination and alkaline phosphatase activity. To our knowledge, this is the first study to evaluate the microbiological quality of raw milk and dairy products in China. We hope this work could attract the public attention of dairy products quality and safety issue. In a word, thank you very much again for sharing your previous time to assess our work and give us helpful comments.

Best regards,

Zixin Peng, Ph.D/Professor
Research Unit of Food Safety, Chinese Academy of Medical Science (2019RU014), NHC Key Lab of Food Safety Risk Assessment

No.37, Guangqu Road, Chaoyang District, Beijing 100022, P. R. China

Reviewer 2 Report

Dear editor,

The submitted manuscript describes sampling of raw, pasteurized and sterilized (UHT) milk for the analysis of alkaline phosphatase activity, aerobic plate counts, mesophilic Bacillus counts and thermophilic Bacillus counts and aims to provide a correlation analysis of alkaline phosphatase activity to the presence of microorganisms. Although the title indicates that this work is primarily a “correlation analysis”, it does not add much to the scientific knowledge towards understanding these correlations. The main idea behind measuring ALP activity is that the enzyme thermal inactivation kinetics would be similar to some heat-resistant microorganisms present in raw milk, therefore, an indirect indicator, rather than having a correlation to the presence or levels of microorganisms. Therefore, I think the approach followed here should be carefully revised.

Besides the ALP activity, the authors report the concentration of aerobic bacteria and Bacillus species in raw, pasteurized and UHT milk and it might be important for risk assessment and baseline contamination research by the Chinese authorities. However, it is worth noting that the methods are not described properly, which makes the results harder to understand.

Therefore, I cannot recommend the acceptance of this manuscript in the current form.

Regards.

Introduction:

Last paragraph: It is appropriate to give the aims of the study as (1) and (2) but this notation can cause confusion with the in-text references. Either change the references to [#] format or find another notation for the aims of the study.

Methods:

Section 2.1: Any sampling plan in particular? Or random or convenience sampling? If random, please also indicate the method for random number generation.

Section 2.2:

Please explain the methods briefly to save the reader from searching through the references which may not be readily available in English such as Chinese Standards. Also for the reference Lücking et al., no enumeration methods were included in the original cited paper and they used a bunch of different methods. Therefore, please explain the methods briefly (but in enough detail so that it’s replicable) in your paper. Same for the enzyme activity assay. It is essential to report the method so that the U unit would make sense.

Another important issue is the limits of detection and quantification in your methods of analysis. Please indicate LOD and LOQ in the manuscript and show the calculations when you respond to this reviewer’s comment. In your results table, I see values as low as 0.014 log10CFU/ml (1.03 CFU/ml) and I wonder what would be this method? I can use it in the future. Most plate count methods will have LOQ higher than 1 or 2 log10CFU/ml and it is not appropriate to do any statistical analysis below LOQ or LOQ of the selected method.

Section 2.3: Indicate “SPSS” in the text.

What correlation? Spearman, Pearson or any other method? The selection of the method is particularly important. As the result tables indicate, your data is not normally distributed, therefore, you cannot use any method that requires normality and in most cases, you will need some transformations to ensure normality.

Any hypothesis testing for the calculated correlation coefficiences? (p-values)

You can give a proper citation to the mentioned website.

Results section:

“1. Results” not accurate. Check numbering of headings and subheadings.

The mean value of the aerobic Bacillus count in raw milk was ten times higher than that of the thermophilic aerobic Bacillus count.”: This statement is not correct because the counts were log transformed. On the absolute scale, the actual difference is about 7 times.

“raw milk samples contained >500000 mU/L alkaline phosphatase activity; however, the mean and median values of alkaline phosphatase ac-tivity were slightly below this value”: Any statistically significant difference for this value and others as well?

“The detection rate of the aerobic plate count was 4.05% (25/617).” You mention ‘detection’ here, so again it is important to state your detection limit. It is not appropriate to report detection without any enrichment step, especially if you are sampling a large volume such as 500 ml.

If only 4.05% of the samples were positives, did you just averaged everything and considered negative samples as zero? This should not be done. You should only report the average from the positive samples and indicate that the other samples were negative. If you plot every data on a histogram, you will see that you have a zero-inflated distribution, not a normal distribution and means and standard deviations will not be valid.

Table 1:

 On some rows, you report mean smaller than 1 (e.g. 0.056), however, other values such as median and mean were reported as “<1”, if you can report 0.056, why would you report others as smaller than 1? Also I repeat my concern here that these values are too small, they are probably below your LOQ and LOD. It needs an explanation of how you handled counts below LOQ and LOD. It will be useful if you can make the raw data publicly available.

This table and other tables: Bacillus needs to be italicized.

Footnote: Please cite the documents properly in the references list. “AB” not defined.

Table 2: Same concerns as Table 1. “ABC” not defined.

Figure 1: Alkaline phosphatase activity is irrelevant in raw milk to make any correlations with the microbial loads. It is only an indicator of temperature and time of the heat treatment, not an indicator of microorganisms present.

Discussion:

Can you give a specific example of subsequent processing procedure that can amplify the contamination? Most processess will aim to reduce the number of microorganisms.

“Destroy” is a strong statement for pasteurization. It does not aim to sterilize the food but reduce the number of microorganisms to safer levels to increase the shelf-life and safe-life.

Again, in order to claim that only 4% was contaminated, you need to use a method with very low LOD, samples must be enriched before plating.

How did you determine that the most microorganisms that you detected were heat-resistant? This was not mentioned in the methods.

Are your methods comparable to the Tunisian study so that you can also compare the results?

“Enzyme contamination”: It should not be called contamination as the enzyme is already present in the raw material. It is just an indicator of proper thermal heat treatment. In general enzymes have greater resistance to heat, than the microorganisms, so it is just a quicker way to confirm that the milk is heated and kept at an appropriate temperature. Therefore, I suggest you discuss the thermal inactivation kinetics of the alkaline phosphatase versus the thermal inactivation kinetics of the microorganisms studied here, especially the most heat resistant thermophilic Bacillus strains. In your reference #9, it is stated that the thermal denaturation kinetics of ALP is similar to heat-resistant pathogens, therefore comparison with non-resistant microorganisms does not make sense.

Discussion of correlation is not enough, as the title of this manuscript mentions the “correlations”. I would expect the whole paper would be about discussing these correlations. The discussion provided here is short of explaining the weak correlations. As said before, this enzyme has a greater heat resistance than most of the aerobic bacteria. Therefore, bacteria will inactivate quicker than the enzyme and you will not be able to detect statistical correlations. Until the enzyme is denaturated, most of the APCs will already be inactivated. Thermophilic Bacilli are more resistant, therefore you were able to enumerate some, in the samples that showed high enzymatic activity. Therefore, this is not due to the complex composition of raw milk and influence of a larger number of microorganisms. Overall, direct correlation might not be an optimum analysis for this study.

Pasteurization is not enough for complete elimination of microorganisms from milk. Therefore, “properly pasteurized” does not mean “free of pathogens”

You mentioned “relatively high positive correlation”. I would call R=0.8 to be a “relatively high correlation”, someone else may have different cutoff. Therefore, it is really important to report and discuss if these correlations were significant. 0.53 does not seem to be a high correlation.

 In the discussion section, I would expect an explanation of practical applications of this project. Do you suggest alkaline phosphatase activity is not a good indicator and it should not be used anymore? Any alternatives, and their effects on public health and economic risks.

Author Response

Reviewer 2

Comments and Suggestions for Authors

Dear editor,

The submitted manuscript describes sampling of raw, pasteurized and sterilized (UHT) milk for the analysis of alkaline phosphatase activity, aerobic plate counts, mesophilic Bacillus counts and thermophilic Bacillus counts and aims to provide a correlation analysis of alkaline phosphatase activity to the presence of microorganisms. Although the title indicates that this work is primarily a “correlation analysis”, it does not add much to the scientific knowledge towards understanding these correlations. The main idea behind measuring ALP activity is that the enzyme thermal inactivation kinetics would be similar to some heat-resistant microorganisms present in raw milk, therefore, an indirect indicator, rather than having a correlation to the presence or levels of microorganisms. Therefore, I think the approach followed here should be carefully revised.

Besides the ALP activity, the authors report the concentration of aerobic bacteria and Bacillus species in raw, pasteurized and UHT milk and it might be important for risk assessment and baseline contamination research by the Chinese authorities. However, it is worth noting that the methods are not described properly, which makes the results harder to understand.

Therefore, I cannot recommend the acceptance of this manuscript in the current form.

Regards.

Dear Reviewer,

    Please accept our sincerely many thanks for sharing your precious time to give us valuable comments and suggestions! All your comments and suggests are very important and improved for this work. We have learned much from you. This manuscript has great improved under your help. Your reviews not only help us to revise the manuscript but also give us a guidance for our future work. Many thanks for sharing your valuable knowledge for us!

    We have studied your each comment carefully, and have revised mainly as: (1) descripted the microbiological analyses and enzymatic activity assays briefly in the method section and showed the LODs or LOQ of methods; (2) redid all the statistical analysis, especially on the mean values of microbial contamination and correlation analysis; (3) corrected the misconceptions and misnomer for the alkaline phosphatase activity and pasteurized milk; (4) enriched the discussion section about the alkaline phosphatase activity and correlation analysis of microbial load and alkaline phosphatase activity; (5) corrected the errors you pointed out.

    We prepared a point-by-point reply for all the comments and suggestions. The revised places were highlighted with yellow color. We appreciate for your warm work earnestly, and hope that the correction will meet with your approval.

Best regards,

Yours, Zixin

Research Unit of Food Safety, Chinese Academy of Medical Science (2019RU014), NHC Key Lab of Food Safety Risk Assessment

No.37, Guangqu Road, Chaoyang District, Beijing 100022, P. R. China

Introduction:

Last paragraph: It is appropriate to give the aims of the study as (1) and (2) but this notation can cause confusion with the in-text references. Either change the references to [#] format or find another notation for the aims of the study.

Reply: Thank you very much for your comment. I am very sorry for the mistake. It has been revised into [â… ] and [â…¡] in Line 70 and 72.

Methods:

Section 2.1: Any sampling plan in particular? Or random or convenience sampling? If random, please also indicate the method for random number generation.

Reply: Thanks for the comments. All the samples were collected randomly. One province collected 105~189 samples, covering raw milk, pasteurized milk, and sterilized milk samples. It has been revised in Line 81-84. We also added a supplementary Table to show the sampling amount in each province.

Section 2.2:

Please explain the methods briefly to save the reader from searching through the references which may not be readily available in English such as Chinese Standards. Also for the reference Lücking et al., no enumeration methods were included in the original cited paper and they used a bunch of different methods. Therefore, please explain the methods briefly (but in enough detail so that it’s replicable) in your paper. Same for the enzyme activity assay. It is essential to report the method so that the U unit would make sense.

Reply: Thank you very much for this precious comment. I am sorry for the unclear methods introduction. The aerobic plate count determination, aerobic Bacillus and thermophilic aerobic Bacillus count method, and alkaline phosphatase activity assay method were added in Line 92-100, 104-127, and 129-140, respectively. The reference of aerobic Bacillus and thermophilic aerobic Bacillus were changed into NEN 6813:2014 and NEN 6809:2014. The reference of alkaline phosphatase activity assay has been changed into “Albillos SM, Reddy R, Salter R. Evaluation of alkaline phosphatase detection in dairy products using a modified rapid chemiluminescent method and official methods. J Food Prot. 2011;74(7):1144-54.” with more details.

Another important issue is the limits of detection and quantification in your methods of analysis. Please indicate LOD and LOQ in the manuscript and show the calculations when you respond to this reviewer’s comment. In your results table, I see values as low as 0.014 log10CFU/ml (1.03 CFU/ml) and I wonder what would be this method? I can use it in the future. Most plate count methods will have LOQ higher than 1 or 2 log10CFU/ml and it is not appropriate to do any statistical analysis below LOQ or LOQ of the selected method.

Reply: Many thanks for the precious comment! This point is very important. We have added the LOD (1 Log10 CFU/mL) of aerobic plate count determination, aerobic Bacillus and thermophilic aerobic Bacillus count method. For the alkaline phosphatase activity assay, the LOD is 1.30 Log10 mU/L (20 mU/L) and the LOQ is 1.78 Log10 mU/L (60 mU/L).

According to the ISO 16140-2:2016 Microbiology of the food chain - Method validation - Part 2: Protocol for the validation of alternative (proprietary) methods against a reference method, the LOQ of microbial count don’t need to be determined for counting visible colonies of the target microorganism. The LOQ is only relevant when the measurement principle of the alternative method is not based on counting visible colonies of the target microorganism and shall therefore be determined in these cases (ISO 16140-2:2016, 6.1.4.1). Therefore, we didn’t show the LOQ of the aerobic plate count determination, aerobic Bacillus and thermophilic aerobic Bacillus count method.

Since many samples were negative for the target microorganism, we used 0 to substitute the value (<1 Log10 CFU/mL), some mean values were <1 Log10 CFU/mL in the previous version. We are glad to learn that it is not appropriate to do any statistical analysis below LOD of the selected method, therefore, we only report the average from the positive samples and added a footnote for Table 1 and 2 “The mean±SD only determined the value ≥LOD for APC, AB, and TAB or ≥LOQ for APA.” and “The mean±SD only determined the value ≥LOD for APC, AB, and TAB.” for the instruction.

Section 2.3: Indicate “SPSS” in the text.

Reply: It has been added in Line 144.

What correlation? Spearman, Pearson or any other method? The selection of the method is particularly important. As the result tables indicate, your data is not normally distributed, therefore, you cannot use any method that requires normality and in most cases, you will need some transformations to ensure normality.

Reply: Thank you so much for this comment. Our data is not normally distributed so we should use Spearman Correlation but not the Pearson Correlation. Thus, we use the Spearman Correlation to re-analysis all the data. The revised method is shown in Line 145-149.

Any hypothesis testing for the calculated correlation coefficiences? (p-values)

Reply: Thanks for the comments. The p value is very important for the correlation analysis. This has been supplemented in Line 146.

You can give a proper citation to the mentioned website.

Reply: It has been added in Line 151.

Results section:

“1. Results” not accurate. Check numbering of headings and subheadings.

Reply: I am sorry for the mistake. It has been revised.

“The mean value of the aerobic Bacillus count in raw milk was ten times higher than that of the thermophilic aerobic Bacillus count.”: This statement is not correct because the counts were log transformed. On the absolute scale, the actual difference is about 7 times.

Reply: Thank you for correction. I am sorry for the mistake. This sentence has been deleted since we only report the average of aerobic Bacillus count and thermophilic aerobic Bacillus count from the positive samples.

“raw milk samples contained >500000 mU/L alkaline phosphatase activity; however, the mean and median values of alkaline phosphatase ac-tivity were slightly below this value”: Any statistically significant difference for this value and others as well?

Reply: This sentence has been deleted since we recounted the mean value.

“The detection rate of the aerobic plate count was 4.05% (25/617).” You mention ‘detection’ here, so again it is important to state your detection limit. It is not appropriate to report detection without any enrichment step, especially if you are sampling a large volume such as 500 ml.

Reply: Thank you for the comments. This comment is very important. The LOD of aerobic plate count is 1 Log10 CFU/mL. This sentence has been revised into “The proportion of aerobic Bacillus was 4.05% (25/617).” in Line 193.

If only 4.05% of the samples were positives, did you just averaged everything and considered negative samples as zero? This should not be done. You should only report the average from the positive samples and indicate that the other samples were negative. If you plot every data on a histogram, you will see that you have a zero-inflated distribution, not a normal distribution and means and standard deviations will not be valid.

Reply: Many thanks for the comments. As you said, I considered negative samples as zero and just averaged everything in the previous comment. In the revised version, I have followed your comment that only report the average from the positive samples.

Table 1:

On some rows, you report mean smaller than 1 (e.g. 0.056), however, other values such as median and mean were reported as “<1”, if you can report 0.056, why would you report others as smaller than 1? Also I repeat my concern here that these values are too small, they are probably below your LOQ and LOD. It needs an explanation of how you handled counts below LOQ and LOD. It will be useful if you can make the raw data publicly available.

Reply: Thank you for the comment. In the previous version, I considered negative samples as zero and just averaged everything in the previous comment, so the mean value was so low as smaller than 1. In the revised version, I have followed your comment that only report the average values from the positive samples.

This table and other tables: Bacillus needs to be italicized.

Reply: I am sorry for the mistake. It has been revised in Line 159, 160 and 165, 166.

Footnote: Please cite the documents properly in the references list. “AB” not defined (Line 161).

Reply: It has been cited with references (Line 162, 163). “AB” has been defined (Line 161).

Table 2: Same concerns as Table 1. “ABC” not defined.

Reply: All the pointed errors have been revised in Line 165-167.

Figure 1: Alkaline phosphatase activity is irrelevant in raw milk to make any correlations with the microbial loads. It is only an indicator of temperature and time of the heat treatment, not an indicator of microorganisms present.

Reply: Thanks for the comment. This comment has been added in the Discussion part in Line 356-358.

Discussion:

Can you give a specific example of subsequent processing procedure that can amplify the contamination? Most processess will aim to reduce the number of microorganisms.

Reply: Thanks for the comment. This sentence has been revised into “Microbial contamination of raw milk originating from farms can increase spoilage and wastage and adversely affect producers, traders, and consumers” (Line 262-264).

“Destroy” is a strong statement for pasteurization. It does not aim to sterilize the food but reduce the number of microorganisms to safer levels to increase the shelf-life and safe-life.

Reply: It has been revised into “because it reduces most heat-resistant and all other non-spore-forming microbes to safer levels to increase the shelf-life and safe-life” in Line 276-278.

Again, in order to claim that only 4% was contaminated, you need to use a method with very low LOD, samples must be enriched before plating.

Reply: It has been revised into “only 4% of sterilized milk samples showed an aerobic plate count ≥1 log10 CFU/mL.” (Line 298-299).

How did you determine that the most microorganisms that you detected were heat-resistant? This was not mentioned in the methods.

Reply: I am sorry for making the misunderstanding. It has been revised into “A previous study demonstrated that most microorganisms present in sterilized milk were heat-treatment-resistant strains or those that originated from post-sterilization contamination” (Line 299-302).

Are your methods comparable to the Tunisian study so that you can also compare the results?

Reply: Thanks for the comment. We used the different method with the Tunisian study, therefore, the sentence about the compare has been deleted.

“Enzyme contamination”: It should not be called contamination as the enzyme is already present in the raw material. It is just an indicator of proper thermal heat treatment.

Reply: I am sorry for my mistake on the knowledge. Thank you very much for your correction. The sentence has been revised into “In our survey, a total of 36.18% of raw milk samples con-tained >500000 mU/L alkaline phosphatase activity, which indicated widespread active enzyme.” (Line 332-335).

 In general enzymes have greater resistance to heat, than the microorganisms, so it is just a quicker way to confirm that the milk is heated and kept at an appropriate temperature. Therefore, I suggest you discuss the thermal inactivation kinetics of the alkaline phosphatase versus the thermal inactivation kinetics of the microorganisms studied here, especially the most heat resistant thermophilic Bacillus strains. In your reference #9, it is stated that the thermal denaturation kinetics of ALP is similar to heat-resistant pathogens, therefore comparison with non-resistant microorganisms does not make sense.

Reply: Thank you for your comment. This comment is very valuable. It has been revised in Line 342-355.

Discussion of correlation is not enough, as the title of this manuscript mentions the “correlations”. I would expect the whole paper would be about discussing these correlations. The discussion provided here is short of explaining the weak correlations. As said before, this enzyme has a greater heat resistance than most of the aerobic bacteria. Therefore, bacteria will inactivate quicker than the enzyme and you will not be able to detect statistical correlations. Until the enzyme is denaturated, most of the APCs will already be inactivated. Thermophilic Bacilli are more resistant, therefore you were able to enumerate some, in the samples that showed high enzymatic activity. Therefore, this is not due to the complex composition of raw milk and influence of a larger number of microorganisms. Overall, direct correlation might not be an optimum analysis for this study.

Reply: Thank you very much for your comment. I am sorry for letting you disappoint. We redid the correlation analysis using the Spearman method and acquired different results. We revised the results and discussion about the correlation results substantially. We used some your comments directly in the Discussion (Line 342-371). We have learned much from your knowledge. Thank you very much for sharing your precious knowledge with us.

Pasteurization is not enough for complete elimination of microorganisms from milk. Therefore, “properly pasteurized” does not mean “free of pathogens”

Reply: Many thanks for correction. It has been deleted in Line 67 and 329.

You mentioned “relatively high positive correlation”. I would call R=0.8 to be a “relatively high correlation”, someone else may have different cutoff. Therefore, it is really important to report and discuss if these correlations were significant. 0.53 does not seem to be a high correlation.

Reply: It has been deleted and redid with the Spearman analysis.

In the discussion section, I would expect an explanation of practical applications of this project. Do you suggest alkaline phosphatase activity is not a good indicator and it should not be used anymore? Any alternatives, and their effects on public health and economic risks.

Reply: Thanks for the comment. To our knowledge, this is the first study to evaluate the microbiological quality of raw milk and dairy products from 13 major dairy production and consumption provinces in China. From the results, we have an overall survey data for the microbial contamination and alkaline phosphatase activity in raw milk and dairy products. These results facilitate the awareness of public health safety issues and the involvement of dairy product regulatory agencies in China. Moreover, from this study we also improved our understanding about the correlation be-tween microbial contamination and alkaline phosphatase activity. Our results showed that the alkaline phosphatase activity was a good indictor for the efficacy of thermal pasteurization. All above was shown in the discussion and conclusion part.

Reviewer 3 Report

The manuscript submitted for review "A new method for detecting ALP in pasteurized milk by the dairy industry" may be a useful application study. Despite valuable observations and impressive research material, the work needs to be supplemented. Below are the most important suggestions for necessary changes:

the lack of line numbers hinders the legibility of the review

Introduction: "Alkaline phosphatase activity has been used to confirm pasteurization in dairy products" - needs to be corrected, illogical

Materials and methods: The research material was obtained from 13 regions in China. Information on the conditions of transport and storage of samples prior to analysis is necessary. The Chinese standard for PCA analysis GB 4789.2-2016 is mentioned only in the methodology, the authors do not refer to the document in the Results and Discussion chapters. There are no standards used to carry out microbiological analyzes in the literature list. Instead, the authors cited publications (9-11). For the methodology for the analysis of alkaline phasphatase enzymatic activity, there is no sub-standard of international scope ISO 11816-1Milk and milk products — Determination of alkaline phosphatase activity — Part 1: Fluorimetric method for milk and milk-based drinks Abbreviations of the analyzes performed should be provided in the methodology (they are provided only under table 1)

There is no basic information on the legitimacy of using the linear Pearson correlation coefficient (parametric test), i.e. a test checking the normal distribution of numerical data. The results presented in Table 1 raise reservations. High SD values were obtained, exceeding the average values (APC - past, AB and TAB). Values described in the table as MAX are not possible. In the case of pasteurized milk, the APC value is identical to the value obtained in raw milk (APC and AB). In addition, the low values obtained in the TAB and AB analyzes do not correspond to the reported maximum values.

Bibliography. 10 of the 25 items in the reference list are older than 10 years. Cited items require verification of journal abbreviations

Author Response

Reviewer 3

Dear Reviewer,

Thank you very much for your comments concerning our manuscript. Your kind comments are all valuable and very helpful for revising and improving our manuscript, as well as give important guiding significance to our future study. We have studied each comment carefully and have made correction which we hope to meet with your approval. In the revised version, we descripted the microbiological analyses and enzymatic activity assay methods and redid all the statistical analysis.

We also revised the MS followed your below comments and prepared a reply point-by-point.

We appreciate for your warm work earnestly, and hope that the correction will meet with your approval.

Best regards,

Yours sincerely,

Zixin

Research Unit of Food Safety, Chinese Academy of Medical Science (2019RU014), NHC Key Lab of Food Safety Risk Assessment

No.37, Guangqu Road, Chaoyang District, Beijing 100022, P. R. China

Comments and Suggestions for Authors

The manuscript submitted for review "A new method for detecting ALP in pasteurized milk by the dairy industry" may be a useful application study. Despite valuable observations and impressive research material, the work needs to be supplemented.

R: Thank you very much for your kind approbation. The output and consumption of dairy products in China have increased greatly in recent decades. Chinese consumers have expressed increased demand for safe and healthy dairy products. We hope this work could promote the awareness of public health safety issues.

Below are the most important suggestions for necessary changes:

the lack of line numbers hinders the legibility of the review

R: I am very sorry for the mistake. This mistake gave you much problems to review this manuscript. I have added the line numbers in the revised manuscript.

Introduction: "Alkaline phosphatase activity has been used to confirm pasteurization in dairy products" - needs to be corrected, illogical

R: It has been revised as “Alkaline phosphatase activity has been used to confirm the efficacy of pasteurization in dairy products.” in Line 66-67.

Materials and methods: The research material was obtained from 13 regions in China. Information on the conditions of transport and storage of samples prior to analysis is necessary.

R: Thank you for the comment. We agree with the transportation and storage condition are very important. However, we did not collect the information for the transportation and storage condition, we just collect raw milk and dairy product samples from farms and markets. We will carry out this work in our future study. This has been supplemented in the Conclusion section in Line 377-378.

The Chinese standard for PCA analysis GB 4789.2-2016 is mentioned only in the methodology, the authors do not refer to the document in the Results and Discussion chapters.

R: Thanks for the comment. We described the GB 4789.2-2016 method briefly in Line 91-100.

There are no standards used to carry out microbiological analyzes in the literature list. Instead, the authors cited publications (9-11). For the methodology for the analysis of alkaline phasphatase enzymatic activity, there is no sub-standard of international scope ISO 11816-1Milk and milk products — Determination of alkaline phosphatase activity — Part 1: Fluorimetric method for milk and milk-based drinks Abbreviations of the analyzes performed should be provided in the methodology (they are provided only under table 1)

R: Thanks for the comment. We descripted the aerobic plate count, the plate counts of aerobic Bacillus and thermophilic aerobic Bacillus, alkaline phosphatase activity assay method briefly in Line 91-140. All the methods and National Standards used were cited under the Table.

There is no basic information on the legitimacy of using the linear Pearson correlation coefficient (parametric test), i.e. a test checking the normal distribution of numerical data. The results presented in Table 1 raise reservations. High SD values were obtained, exceeding the average values (APC - past, AB and TAB). Values described in the table as MAX are not possible. In the case of pasteurized milk, the APC value is identical to the value obtained in raw milk (APC and AB). In addition, the low values obtained in the TAB and AB analyzes do not correspond to the reported maximum values.

R: Thank you very much for your comment. This comment is very important and valuable. Our data is not normal distribution; thus we redid all the statistical analysis in the Table 1 and 2 and Figure 1 and 2. In the previous manuscript, I considered negative samples (<LOD) as zero and just averaged everything. In the revised version, I have only reported the average values from the positive samples (The value >LOD or LOQ).

Bibliography. 10 of the 25 items in the reference list are older than 10 years. Cited items require verification of journal abbreviations

R: Thank you for the comment. The references have been updated to new ones.

Round 2

Reviewer 2 Report

Dear authors,

Thank you for a very thorough revision of your article. 

Regards.